# Should We Regularly Assess Hemoglobin Levels Following Elective Total Knee Arthroplasty, with the Administration of TXA and Without the Usage Tourniquet? An Observational Study from a Single Center

**DOI:** 10.3390/medicina60121964

**Published:** 2024-11-28

**Authors:** Shanny Gur, Mor Bracha Akselrad, David Segal, Yuval Fuchs, Dan Perl, Alon Fainzack, Nissim Ohana, Eyal Yaacobi, Michael Markushevich, Yaron Shraga Brin

**Affiliations:** Orthopedic Surgery Division, Meir Medical Center, Tel Aviv University, Tchernichovsky St. 59, Kfar-Saba 6997801, Israel; moraxelrod@gmail.com (M.B.A.); dudisegal@gmail.com (D.S.); yuvalfuchs@gmail.com (Y.F.); danperl2208@gmail.com (D.P.); pastelico@gmail.com (A.F.); niso62@yahoo.com (N.O.); yaacobi.eyal@gmail.com (E.Y.); michaelmark01@gmail.com (M.M.)

**Keywords:** total knee replacement, blood transfusion, fast-tracking

## Abstract

*Background and Objectives:* The aim of this study was to test whether there is a necessity for routine postoperative laboratory testing in patients undergoing primary elective unilateral total knee arthroplasty (TKA), with the administration of Tranexamic Acid (TXA) and without the use of tourniquet. *Materials and Methods*: This observational, retrospective cohort study was conducted at Meir Medical Center. The data were collected in 2018–2022. Patients were collected in a consecutive manner. There were no exclusion criteria for this study. The inclusion criteria were individuals who underwent elective TKA due to end-stage osteoarthritis (OA). We conducted a multivariable logistic regression analysis to determine the factors associated with postoperative hemoglobin (PoOHb) < 9 g/dL and the optimal cutoff to detect those patients postoperatively. *Results*: A total of 271 patients were included. The mean preoperative hemoglobin (PrOHb) was 13.02 ± 1.42 g/dL, and the mean PoOHb was 11.5 ± 1.34 g/dL. The mean decrease in Hbg levels following surgery was 1.52 ± 0.91 g/dL. In all, 271 patients had a PoOHb level ≥ 9 g/dL or above. A total of 16 patients received packed red blood cells following surgery. Patients with PoOHb < 9 g/dL had a significantly lower PrOHbg. In the current study, all 11 patients who had PoOHb < 9, 10 (90.9%) had PrOHb ≤ 1011.95 55 g/dL, compared to 25 (49.6%) of the 260 patients who had PoOHb ≥ 9. The odds ratio for having PrOHb ≤ 10.9511.55 g/dL if PoOHb < 9 g/dL was 206.6710.4 (95% CI 24.427.164 to 1749.01415.97). A cut-off of 11.2510.25 would have offered a sensitivity of 99% (specificity 93.5%). *Conclusions*: In conclusion, this study supports using selective post-TKR Hgb monitoring rather than adhering to routine practice. An association between postoperative anemia and the need for blood transfusion was found only when the preoperative baseline hemoglobin threshold was less than 10.95 g/dL.

## 1. Introduction

Osteoarthritis (OA) is the most prevalent joint disorder in the United States [1]. Among adults aged 60 and older, about 10% of men and 13% of women experience symptomatic knee OA. The number of individuals affected by symptomatic OA is expected to rise due to the aging population and the growing obesity epidemic [2]. Several factors contribute to the development of knee osteoarthritis, including increasing age, female gender, obesity, a history of knee injury, occupations that involve prolonged kneeling, squatting, or heavy lifting, participation in high-impact sports, genetic predisposition, quadriceps weakness, physical inactivity, certain metabolic disorders, and others [3]. The treatment for knee osteoarthritis can be either conservative or surgical, particularly in cases of end-stage osteoarthritis.

Total knee arthroplasty (TKA) is one of the most frequent and effective surgical procedures for end-stage osteoarthritis [4]. It is one of the most common elective surgeries in modern orthopedic surgery, and its frequency is expected to increase substantially by 2030 [5]. While TKA can alleviate pain and improve function, it is important to be mindful of potential risks, such as perioperative blood loss [6].

Significant blood loss requiring allogeneic blood transfusion remains a concern, with the reported rates ranging from 10% up to 38% [7,8]. However, modern perioperative practices have substantially reduced these rates. Despite these advances, routine blood monitoring following major orthopedic surgeries, such as TKA, is still widely acknowledged as a standard clinical practice [9,10]. Postoperative blood transfusions are not considered benign, and aside from their financial burden, they carry risks for immune and non-immune reactions, including a noticeable increase in joint infection [11]. The use of tranexamic acid (TXA) in patients undergoing TKA has demonstrated efficacy in reducing the number of patients requiring blood transfusions, as well as diminishing the quantity of transfusion units necessary during and after surgery. Additionally, it may contribute to decreased hospital expenses and a shorter length of stay [12].

Improvements Current practices vary due to inconsistent guidelines and varying thresholds for transfusion. This study addresses the gap in understanding which patients truly benefit from intensified monitoring in both modern surgical techniques and perioperative care pathways, and questions the necessity of routine postoperative laboratory tests for patients undergoing primary elective TKA.

## 2. Materials and Methods

### 2.1. Study Design

This observational, retrospective cohort study was conducted at Meir Medical Center, a secondary care academic medical facility in the center of Israel. Patients were collected in a consecutive manner.

### 2.2. Statistical Analysis

We conducted a univariate comparison analysis to detect variables that had a statistically significant association with PoOHb < 9 g/dL. A receiver operating characteristic (ROC) curve analysis was performed, where PrOHb was plotted against PoOHb to determine the optimal cutoff to detect those patients with PoOHb < 9 g/dL. The optimal cut-off was determined by a sensitivity of 90%, which was regarded by the authors as an adequate sensitivity for this test. Since sensitivity was more important than specificity, the Yoden index was not utilized. Since PrOHb was the only factor that was found to have a statistically significant association with the outcome variable, controlling for potential confounders was not required.

Perioperative Management: We typically conduct blood tests up to a week before the operation and on the morning of the first postoperative day. This is standardized for all patients. Preoperative blood transfusion was given to those who had PrOHb < 9 g/dL.

TXA Criteria for Blood Transfusion:

The inclusion criteria for this study were individuals who underwent elective total knee arthroplasty (TKA) with the administration of tranexamic acid (TXA), without the use of a tourniquet, due to end-stage osteoarthritis (OA). Eligible participants also had their available demographic and health-related data, including preoperative hemoglobin (PrOHb) levels, obtained within two weeks prior to surgery and postoperative hemoglobin (PoOHb) levels measured one day after surgery.

TXA was indicated for all patients undergoing TKA, except for those with any of the following contraindications: active thromboembolic disease, severe renal impairment, hypersensitivity to TXA, uncontrolled hypercoagulable states, or a history of seizures.

Data Collection:

Data for this study were collected retrospectively from 2018 to 2022. Information was extracted by three physicians involved in the study, utilizing electronic medical records.

Surgical techniques:

The same two chief surgeons, utilizing similar techniques, led all the cases. These included the following: (1) prophylactic pre-incision single dose IV antibiotics and 1 g of IV TXA, preoperatively. (2) No tourniquets were used at any time of the operation. (3) The medial parapatellar approach was chosen in all cases. (4) Posterior capsule and soft tissues were infiltrated after femoral chamfer cuts with another 60 mL of BAD solution. Local infiltration points included the following: quadriceps mechanism, medial gutter, and pes anserine. (5) Intra-articular lavage was used after capsule closure with 3 g of TXA. (6) Low-molecular-weight heparin (enoxaparin 40 mg) was administered subcutaneously once a day for 14 days postoperatively, starting 12 h after surgery.

Hemoglobin Measurement: A simple complete blood count (CBC) test was performed at the hospital to determine the need for a blood transfusion, Participants were those who underwent TKA.

### 2.3. Ethics

This study was approved by the Ethics Committee located in Meir Medical Center, on April 2022. Informed consent was not necessary due to the retrospective nature of the study, which used deidentified data.

## 3. Results

A total of 271 patients were included in the study, including 189 (69.7%) females, and the mean age was 69 (range 41 to 91) years. Patients were divided into two groups based on their postoperative hemoglobin levels: those with levels below 9 g/dL and those with levels of 9 g/dL or higher. The comorbidity rates were comparable between the two groups (Table 1).

The mean PrOHb was 13.02 ± 1.42 g/dL (range 9.4 to 17.1 g/dL), and the mean PoOHb was 11.5 ± 1.34 g/dL (range 7.8 to 16.1). The mean decrease in Hb levels following surgery was 1.52 ± 0.91 g/dL (range −1.7 to 4.2). In all, 260 (95.9%) patients had a PoOHb level ≥ 9 g/dL or above, and 11 (4.1%) had a PoOHb of 7–9 g/dL (none had levels below 7). A total of 16 (5.9%) patients received packed red blood cells following surgery. We found no differences in the use of spinal anesthesia and the duration of surgery (Table 1).

The postoperative results are illustrated in Table 2. Patients with PoOHb < 9 g/dL had a significantly lower PrOHb (10.29 ± 0.58 g/dL vs. 13.13 ± 1.32, *p* < 0.001). A ROC curve analysis had an area under the curve of 98.2% (95% CI 96.6% to 99.8%, *p* < 0.001, excellent model quality). According to this analysis, to detect patients with a PoOHb < 9 g/dL with a sensitivity of 90% (specificity 99%), all patients with a PrOHb ≤ 11.55 g/dL would have to be tested postoperatively for Hb levels. In the current study, all 11 patients who had a PoOHb < 9 had PrOHb ≤ 11.55 g/dL, compared to 25 (9.6%) of the 260 patients who had PoOHb ≥ 9. The odds ratio for having PrOHb ≤ 11.55 g/dL if PoOHb < 9 g/dL was 10.4 (95% CI 7.164 to 15. 97). A cut-off of 10.25 would have offered a sensitivity of 99% (specificity 93.5%).

## 4. Discussion

The most important finding of the study is that patients were at a higher risk of blood transfusion when their preoperative baseline hemoglobin was less than 10.95 g/dL. Hence, we recommend a routine postoperative blood exam in patients undergoing TKA with TXA and without the use of a tourniquet when their preoperative Hb is below 11 g/dL.

In this single-center, retrospective, cohort study involving 271 patients who underwent elective total knee arthroplasty, we observed that the only baseline variable that had a significant association with postoperative anemia (defined as postoperative Hb < 9 g/dL) was a lower preoperative hemoglobin level (10.3 ± 0.6 vs. 13.1 ± 1.3). While this finding was not unexpected, it underscores that neither sex, age, various comorbidities, nor the duration of surgery had any correlation with the occurrence of postoperative anemia. This finding adds to the growing body of evidence that suggests that the duration of post-TKA observation should be limited and early discharge should be enabled [10].

Preoperative anemia is common, affecting 15% to 30% of patients undergoing elective arthroplasty. It is a notable risk factor for postoperative complications following primary arthroplasty. The severity of anemia is closely linked to the increased odds of experiencing postoperative complications, necessitating vigilant postoperative monitoring, close follow-up, and perioperative blood transfusions [7]. Recent studies have evaluated the risk factors for anemia and blood transfusion: a low preoperative Hb level, higher blood loss during surgery, longer operative time, higher ASA score, the use of surgical drains, female gender, older age, higher BMI, comorbidities like Chronic Obstructive Pulmonary Disease (COPD) and bleeding disorders, black race, and steroid use [13]. Interestingly, our study found that a lower preoperative hemoglobin level was the only factor associated with a lower postoperative hemoglobin level. A study by Bailey et al. found that patients with preoperative anemia had poorer outcomes following TKA. They were four times more likely to require a blood transfusion compared to non-anemic patients. Similarly, those with anemia experienced a 40% higher rate of postoperative complications compared to their non-anemic counterparts. Furthermore, their length of hospital stay was approximately 20% longer than non-anemic patients [14].

In the past decade, advancements in preoperative screening, lab tests, and surgical protocols such as hypotensive anesthesia, tourniquet avoidance, and the use of TXA have significantly reduced blood loss and the need for transfusions during and after surgery [15,16]. These advances have also been associated with a significant reduction in complication, reoperation rates, and the length of hospital stay [8]. In our practice, we opt to use TXA, avoid the use of tourniquets, and maintain controlled hypotension throughout surgery, resulting in a low rate of postoperative anemia. TXA is increasingly employed as a strategy for minimizing perioperative blood loss during TKA [17]. TXA is a synthetic amino acid that inhibits fibrinolysis by using a reversible blockade of lysine binding sites on plasminogen molecules, inhibiting its activation, which prevents plasmin from binding with fibrinogen and fibrin structures after clot formation. In our institution, patients are administered intravenously 1 g of TXA prior to skin incision and 3 g of an intra-articular before wound closure. Several studies have shown that TXA is a cost-effective way of reducing transfusion-related expenses and overall hospitalization costs in the setting of TKA [8,11,18].

A meta-analysis performed by Zhi-Gao Yang et al. showed that the use of TXA for patients undergoing TKA is effective and safe for reducing blood loss, the number of blood transfusion units, and the number of patients needing transfusion. However, it does not change the prevalence of deep vein thrombosis or pulmonary embolism [8].

A meta-analysis by Zhang et al. found that abstaining from tourniquet use in TKA resulted in superior clinical outcomes, particularly in terms of lower complication rates and an improved range of motion during the early postoperative phase. Interestingly, that study found no significant disparity in actual blood loss between the tourniquet and non-tourniquet groups [19].

A systematic review and meta-analysis comprising 14 randomized controlled trials involving 1329 patients indicated that the use of tourniquets in TKA was linked to several adverse outcomes. These included an elevated overall risk of infection, increased intraoperative blood loss, a greater need for blood transfusion, and prolonged hospital stays. Consequently, the findings from this meta-analysis do not advocate for the routine utilization of tourniquets in TKA [20].

In our institution, TKA is performed by a team of three hip and knee arthroplasty surgeons. All TKA procedures included in our study were conducted by these three specialists and were performed without the use of a tourniquet. Generally, we use the Unity KneeTM (Corin, Cirenceter, UK) type of implants for TKA. Several studies dealing with the size and design of implants in total knee arthroplasty (TKA) have shown that they can significantly influence blood loss during and after surgery. Here, various factors come into play: Cruciate-Retaining (CR) vs. Posterior-Stabilized (PS) Implants. CR implants preserve the posterior cruciate ligament, which may result in less bone resection and potentially less blood loss compared to PS implants, which require the resection of the ligament and may lead to increased bleeding due to greater soft tissue disruption [21]. We used the same type of implant for all patients; thus, we have no statistics to show for other options. We suggest that blood monitoring is performed in all patients with a baseline Hbg of less than 10.95 g/dL, as well as patients who experience bleeding complications during surgery.

The consistently low rate of postoperative anemia in our cohort can also be attributed to the fact that all surgeries were performed by the same experienced team. Research has shown that higher surgeon and hospital procedure volumes are typically linked to better outcomes in TKA. One explanation for this is a shorter operating time. Longer surgeries may result in greater blood loss, which can increase the risk of postoperative anemia [22]. 

In the context of advancing rapid recovery after TKA, the necessity for routine postoperative blood tests for patients who undergo primary, elective TKA should be reassessed [17]. Although potentially valuable, these tests could lead to discharge delays and might increase complications such as delirium, infections, mortality, and, of course, have an impact on hospitalization costs [23].

In a large institutional registry including 522 patients who underwent primary TKA, only three individuals (0.5%) required postoperative blood transfusion. Among these cases, two patients did not have anemia at baseline but did not receive TXA, and although the third patient received TXA, their preoperative hemoglobin was 10.1 g/dL, falling below our suggested threshold.

Candidates for fast-tracking TKA are typically medically optimized patients who would benefit from an accelerated recovery pathway. Some fast-track programs aim for a 24 h stay or even same-day discharge when certain criteria are met [24].

The present study adds to the growing evidence that postoperative Hb monitoring is not necessary for most patients after primary elective TKA and shows that most of these patients can be on a “fast-track” discharge schedule after surgery, without needing to undergo postoperative blood tests via additional hospitalization.

Our results suggest that patients with preoperative Hg > 11.55 g/dL and no intraoperative complications do not need routine postoperative blood tests following TKA and that fast-track TKA can be applied. In these cases, a specialized outpatient team could conduct the routine post-operative blood tests at the patient’s home. It is already a standard practice to reserve blood transfusions for cases where Hb levels fall below 7 g/dL, or below 8 g/dL in special circumstances. In this study, however, we selected a postoperative Hb level of 9 g/dL as the threshold for intervention. 

It is generally recommended that blood transfusions are administered when hemoglobin (Hb) levels fall below 8 g/dL [25]. Additionally, Hb levels are known to decline gradually, reaching their lowest point around the fifth day after surgery. In our study, we focused on treating elderly patients with multiple comorbidities, many of whom have metabolic syndrome, distinguishing them from healthier, younger patients. To facilitate safe discharge without the need for post-discharge blood tests, we aimed to identify patients at potential risk after five days without medical supervision. Consequently, we established a cutoff Hb level of 9 g/dL for our analysis. Our findings aim to assess patients who may be at increased risk. Rather than recommending transfusions for these individuals, we advocate for closer monitoring to ensure their safety.

Regarding concerns about active hemorrhage, we ensured that all cases were thoroughly evaluated and controlled at the end of the operation.

We assumed that in this elderly cohort, Hb levels would continue to decline after the first postoperative day and would reach significantly lower levels in the following days. Since our goal was to evaluate the feasibility of fast-tracking, we chose a higher post-operative Hb cutoff to ensure a safer approach for patients who were expected to be discharged to their homes on the same day as surgery.

Fast-track TKA is an approach that aims to optimize patient outcomes and accelerate recovery following surgery. There are several advantages to this protocol, mainly decreasing the duration of hospital stay and the prevalence of complications attributed to longer hospitalization, such as delirium, deep vein thrombosis, pulmonary embolism, surgical site infections, urinary tract infections, pneumonia and others [21]. This also has benefits such as cost savings, mainly for healthcare providers. This study had several limitations. A post hoc power analysis with an alpha of 5% and power of 80% showed that, for continuous variables, a difference of 1.14 g/dL in preoperative hemoglobin could be detected with statistical significance, and for categorical variables, differences of 0.4 in proportions (0.5 vs. 0.9) could be detected with statistical significance. This limited the study’s ability to detect differences and was acknowledged as a substantial weakness. The surgeries were conducted by three attending surgeons who collaborate in a single public hospital. This limits the generalizability of the findings. There was no strict protocol that determined the criteria for a blood transfusion. The data on patients’ BMI were partial, which limited our ability to investigate it as a possible confounder. Inherent limitations, such as potential selection bias and unmeasured confounding factors, should be explicitly acknowledged. Considering these limitations, a prospective, multi-center study with a strict protocol that evaluates the value of postoperative blood testing is needed.

## 5. Conclusions

This study recommends selective hemoglobin monitoring after TKA with the administration of TXA and without the use of tourniquet, instead of routine checks. Anemia post-surgery correlated with transfusion needs only when patients’ preoperative hemoglobin was below 11.55 g/dL. Therefore, we recommend standard postoperative blood monitoring only for these patients. Additional randomized studies with longer follow-up periods are necessary to validate these findings.

## Figures and Tables

**Table 1 medicina-60-01964-t001:** Baseline patient characteristics.

Variable	PoOHb ≥ 9 (n = 260)	PoOHb < 9 (n = 11)	Total (n = 271)	*p* Value
Female, n (%)	179 (68.8%)	10 (90.9%)	189 (69.7%)	0.119
Age (average ± SD)	68.9 ± 8.67	67.82 ± 10.439	68.86 ± 8.7	0.687
BMI (n = 199 available)	31.15 ± 5.42 (n = 190)	29.99 ± 5.07 (n = 9)	31.1 ± 5.39	0.529
Hypertension	140 (53.8%)	7 (63.6%)	147 (54.2%)	0.759
Dyslipidemia	116 (44.6%)	3 (27.3%)	119 (43.9%)	0.357
Diabetes	91 (35%)	2 (18.2%)	93 (34.3%)	0.341
Respiratory disease	23 (8.8%)	0 (0%)	23 (8.5%)	0.607
APLA/DVT/SLE/F5	1 (0.4%)	0 (0%)	1 (0.4%)	1
CVA/TIA/PVD	17 (6.5%)	0 (0%)	17 (6.3%)	1
Cardiac disease	45 (17.3%)	2 (18.2%)	47 (17.3%)	1
Smoker	31 (11.9%)	0 (0%)	31 (11.4%)	0.621

PoOHb = Postoperative hemoglobin, SD = standard deviation, BMI = body mass index, APLA = antiphospholipid antibodies, DVT = deep venous thrombosis, SLE = systemic lupus erythematosus, CVA = cerebrovascular accident, TIA = transient ischemic attack, PVD = peripheral blood vessels.

**Table 2 medicina-60-01964-t002:** Operative and postoperative data.

Variable	Post-op Hb > 9 (n = 260)	Post-op Hb < 9 (n = 11)	Total (n = 271)	*p* Value
TXA	all	all		
Tourniquet use	none	none		
Spinal anesthesia	147 (56.5%)	6 (54.5%)	153 (56.5%)	1
Surgical duration (minutes)	108.99 ± 39.21	104.82 ± 19.359	108.82 ± 38.59	0.726
Hospital Stay (days)	5.63 ± 2.12	5.45 ± 0.82	5.62 ± 2.09	0.784
Preoperative Hb, g/dL	13.132 ± 1.324	10.291 ± 0.582	13.01 ± 1.42	<0.001
Postoperative Hb, g/dL	11.63 ± 1.21	8.42 ± 0.33	11.5 ± 1.34	<0.001
Delta Hb, g/dL	1.5 ± 0.92	1.87 ± 0.49	1.51 ± 0.91	0.189

PoOHb = Postoperative hemoglobin, TXA = tranexamic acid, HB = hemoglobin.

## Data Availability

Data for this study were collected retrospectively. Information was extracted from electronic medical records.

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
