# Peer review of "Should We Regularly Assess Hemoglobin Levels Following Elective Total Knee Arthroplasty, with the Administration of TXA and Without the Usage Tourniquet? An Observational Study from a Single Center"

_medicina, 2024, doi:10.3390/medicina60121964_

Round 1
Reviewer 1 Report
Comments and Suggestions for Authors
Dear Authors,
In this retrospective study is highlighted an interesting topic with both medical and socioeconomic aspects, the necessity of routine evaluation of haemoglobin levels after TKA. On the whole is a well-conducted study, but few points need further clarification.
Data Analysis Paragraph: Please clarify and provide the rationale of using 9g/dL Haemoglobin as a cut-off point. Provide reference. The same issue needs further clarification in the last paragraph of the manuscript; it is mandatory to be clarified that an hypovolemic unstable patient needs resuscitation with fluids and blood. However, hypovolemic shock diagnosis has nothing to do with haemoglobin levels (blood pressure, pulse, urine production per hour, mind status etc). So, please, provide to the readers reference according to them blood transfusion are done in your centre. Last but not least please provide rationale according to , Hb levels will continue to decline after the first postoperative day; for how many days patient will be haemodynamically unstable? Is there any active haemorrhage?
Last paragraph : Please mention in short what is "fast-tracking" procedure? How many days does hospitalisation last? Which patients are appropriate candidates for this procedure?
Please provide data regarding implants and manufacturer. The surgical team was the same, what about implants? As far as I am considered, implant size and design plays a major role in blood loss - CR versus PS dosing, "box" osteotomy, hinged or semi-constrained, bone loss and also the presence of synovitis could increase blood loss. Please update the existed tables with this data and proceed with relative statistical analysis.
Please provide your suggestion according to study results regarding postoperative Hb levels monitoring. Which tests are mandatory and for how many postoperative days? This could form an easily used management algorithm for any recon-orthopaedic surgeon.
Reviewer 2 Report
Comments and Suggestions for Authors
Dear Authors,
Before addressing the scientific content, I would like to highlight some concerns regarding the structure and formatting of your manuscript. The current version does not fully comply with the journal's formatting and structural standards. If your manuscript is selected for the next round of reviews, based on the decision of the editors and other reviewers, I recommend thoroughly revising it according to the journal's official template. Please ensure that the revision follows all the instructions for authors, and includes properly formatted sections, such as the abstract, introduction, methods, results, discussion, and references (from start to finish).
Abstract
I did not receive an abstract in the current version of the manuscript, so I am unable to provide comments on its quality.
Introduction
The introduction should briefly discuss the current standard of care and practices for postoperative hemoglobin monitoring in TKA patients. It would also be beneficial to clearly state why current clinical practices vary and what specific gap this study aims to address. The rationale for focusing on postoperative hemoglobin monitoring requires further elaboration, particularly regarding why the 9 g/dL cutoff was chosen instead of other thresholds commonly used in similar studies. How does this study aim to fill the existing gaps in the literature?
Materials and Methods
I have several concerns regarding the Materials and Methods section. I recommend reorganizing this section to make it more comprehensive and cohesive. You may benefit from dividing it into subsections such as Study Design, Statistical Analysis, Perioperative Management, Transfusion Criteria, Data Collection, Surgical Technique, Hemoglobin Measurement, Outcome Measures, and Participants. The retrospective cohort design is suitable for the research question, but the inherent limitations, such as potential selection bias and unmeasured confounding factors, should be explicitly acknowledged.
The date of ethics committee approval is missing, along with the name of the committee that granted it. Including these details is crucial to verify the ethical oversight of the study.
The sample size justification is lacking. There is no information on power calculation or whether the sample size had sufficient statistical power to detect meaningful differences. Please address this aspect to strengthen the validity of your conclusions.
The criteria for participant inclusion are vaguely defined. It remains unclear whether any patients were excluded based on conditions like pre-existing anemia or other hematological disorders. Please include more specific criteria, such as age range, specific diagnostic criteria for osteoarthritis, and any exclusion criteria that were applied.
Information on how data was collected between 2018 and 2022 is absent. Clarify whether data was extracted from electronic medical records or through another standardized method. Were multiple reviewers involved to ensure reliability? This detail is important for assessing the consistency and quality of data collection.
The timing of postoperative hemoglobin measurement is described as one day post-operative, but it is unclear if this was standardized for all patients. Any variation could significantly affect the results, so this should be explicitly addressed.
The statistical analysis section mentions using multivariable logistic regression, but no details are provided regarding which variables were included or how they were chosen. This is critical information for assessing the robustness of the analysis.
The methods used for hemoglobin measurement are also not described in sufficient detail. Please provide the specific blood testing procedure to improve the reproducibility of your results.
There is no information about the criteria used for determining blood transfusion. Since the study focuses on postoperative hemoglobin levels, defining these criteria is essential for interpretation.
Detailed descriptions of the TKA procedure, including the type of implant used and any standardized surgical protocols followed, are missing. Providing this information will enhance the reproducibility of the study.
The use of TXA is mentioned in the discussion, but there is no description of its administration in the methods (e.g., timing, dose, route). Including these details in the perioperative management section will provide a more complete picture.
Clearly define your primary and secondary outcomes. Currently, they are not distinctly stated, which makes it challenging to understand the focus of your analysis.
Regarding the statistical methods, more details are needed about the ROC curve analysis and the variables included in the model, as well as how potential confounders were managed.
Results
The sample size discrepancy between the two groups (260 vs. 11) is valid and should be noted as a limitation, as it affects the statistical power for the smaller group. In Table 2, the sample size for the Post-op Hb >9 group is incorrectly listed as 289 instead of 260 and needs correction. For the ROC analysis, while the AUC of 98.2% is impressive, the confidence intervals are missing and should be included to clarify the precision of the model. The odds ratio of 206.67, with its wide confidence interval (24.42 to 1749.014), reflects uncertainty due to the small sample size, which should be discussed. The missing BMI data (only available for 199 patients) could impact the results, and this limitation needs to be addressed.
Comments on the Quality of English LanguageProofreading required.
Reviewer 3 Report
Comments and Suggestions for Authors
Dear Editor and authors,
I would like to thank you for sending me this manuscript for review.
I read with great interest and curiosity as an orthopedic surgeon.
The manuscript needs major revision.
Here are my concerns:
1) Title: You should mention that you used TXA and did not use a tourniquet.
2) Abstract: I did not see a detailed abstract.
3) Intorduction is OK.
4) Methods: Posthoc power analysis is desperately needed. How is your hemoglobin blood test regime in your Hospital? Identify please. For me, after TKA, I order a hemogram early postop (the operation night), on the first day, second day, third day, and after that for every five days of Hospital stay. How do you recommend this blood sampling regime should be? How is your TXA regime? You should mention it in the methods, not in the discussion. What are the indications, especially contraindications, for TXA? Do you use surgical drains? What are the indications of packed red blood cell use after TKA? Only Hb below 7? What happens if Hb 8 with a heartbeat is over 100?
5) Tables should be more organized.
6) Results OK.
7) Discussion: This is OK, but please start the discussion with “The most important finding(s) of the study was/were….”
8) Conclusion: Please improve and shorten it. It should be striking for the readers.
It would be stunning if you could suggest a postop blood sampling regime after TKA with TXA and without a tourniquet.
Thank you.
Kind regards.
Reviewer 4 Report
Comments and Suggestions for Authors
The authors are making a retrospective analysis over a 5-year period assessing whether there is a need for routine postoperative blood testing for patients undergoing elective TKA. They have shown that the only significant parameter influencing the postop levels of Hb is the preoperative levels of Hb. None of the other factors analyzed including preoperative comorbidities, OR time, age, weight had any influence. The results of the study might be very useful in clinical practice since it might influence the shift from routine HB follow-up to fast-track strategy for these patients, meaning less stress for the patients and lower accost for the system. The manuscript is well written and easy to follow.
There are some minor issues:
Inclusion criteria: Were the cases collected in a consecutive manner? Are there any exclusion criteria?
Table 1 – A legend of the abbreviations used in the table would be useful.
Table 2 - Same as in table 1
Results – The authors do not mention any complications. Weren’t any? How would complications impact the PoHb levels? Also, the matter of complications is a reason for postop clinical and lab work follow-up. It might be worth discussing this aspect.
Discussions – TXA is not the only way of reducing postop anemia. For instance, preoperative iron compounds in elective surgery have been used for the same reasons. What is the author’s opinion on this matter?
Discussions, paragraph 2 – COPD -Abbreviations should be defined when first mentioned in text
Discussion last paragraph, line 12 – There is a typo “othersf”
Discussion – “In our institution, TKA is performed by a team of three hip and knee arthroplasty surgeons” Actually, this is a good point mentioned. The surgical procedure and the way it is performed is a very important factor influencing the postoperative course of the patient, including postop anemia. Following this line, can the authors discuss if there are any important aspects of the surgical technique itself that might be influential over the postop anemia? What is their standard surgical technique? Despite TKA is performed in similar manner all over the world, there are some variations.
References are not formatted in character with the journal style
Round 2
Reviewer 1 Report
Comments and Suggestions for Authors
Dear Authors,
You have adequately addressed both points raised. There is only one part of the revised manuscript (lines 105-127) needs editing as the comprehension is really difficult.
Reviewer 2 Report
Comments and Suggestions for Authors
Dear Authors,
Thank you for your revisions. To improve its suitability for publication, please ensure that the paper is formatted according to the journal's guidelines. I encourage you to review the author instructions carefully and apply them thoroughly to enhance the overall academic cohesion and structure of the manuscript.
Comments on the Quality of English LanguageMinor grammar improvements are still needed.
Reviewer 3 Report
Comments and Suggestions for Authors
There are some typos to correct and need minor revision, but other than that, I recommend publication. Kind regards.
Comments on the Quality of English LanguageThere are some typos to correct and need minor revision, but other than that, I recommend publication. Kind regards.
